# MANIFOLDNET: A DEEP NEURAL NETWORK FOR MANIFOLD-VALUED DATA

## ABSTRACT

Developing deep neural networks (DNNs) for manifold-valued data sets has gained much interest of late in the deep learning research community. Examples of manifold-valued data include data from omnidirectional cameras on automobiles, drones etc., diffusion magnetic resonance imaging, elastography and others. In this paper, we present a novel theoretical framework for DNNs to cope with manifold-valued data inputs. In doing this generalization, we draw parallels to the widely popular convolutional neural networks (CNNs). We call our network the ManifoldNet.

As in vector spaces where convolutions are equivalent to computing the weighted mean of functions, an analogous definition for manifold-valued data can be constructed involving the computation of the weighted Fréchet Mean (wFM). To this end, we present a provably convergent recursive computation of the wFM of the given data, where the weights makeup the convolution mask, to be learned. Further, we prove that the proposed wFM layer achieves a contraction mapping and hence the ManifoldNet does not need the additional non-linear ReLU unit used in standard CNNs. Operations such as pooling in traditional CNN are no longer necessary in this setting since wFM is already a pooling type operation. Analogous to the equivariance of convolution in Euclidean space to translations, we prove that the wFM is equivariant to the action of the group of isometries admitted by the Riemannian manifold on which the data reside. This equivariance property facilitates weight sharing within the network. We present experiments, using the ManifoldNet framework, to achieve video classification and image reconstruction using an auto-encoder+decoder setting. Experimental results demonstrate the efficacy of ManifoldNet in the context of classification and reconstruction accuracy.

## 1 INTRODUCTION

Convolutional neural networks (CNNs) have attracted enormous attention in the past decade due to their significant success in Computer Vision, Speech Analysis and other fields. CNNs were pioneered by LeCun et al. (1998) and gained much popularity ever since their significant success on Imagenet data reported in Krizhevsky et al. (2012). CNNs have traditionally been restricted to dealing with data residing in vector spaces. However, in the past few years, there is growing interest in generalizing the CNNs and deep networks in general to data that reside on smooth non-Euclidean spaces. In this context, at the outset, it would be useful to categorize problems into 1) those that involve data as samples of real-valued functions defined on a manifold and 2) those that are simply manifold-valued and hence are sample points on a manifold.

In the context of input data being samples of functions defined on smooth manifolds, recently there has been a flurry of activity in developing methods that can cope specifically with samples of functions on a sphere that are encountered in many applications such as, omnidirectional cameras on drones, robots etc., meteorological data and many others. The key property that allows learned weight sharing in CNNs is the equivariance to translations. The simplest technique to achieve equivariance is via data augmentation (Krizhevsky et al., 2012; Dieleman et al., 2015). Cascade of wavelet transforms to achieve equivariance was shown in Bruna & Mallat (2013); Oyallon & Mallat (2015). In Gens (2014), authors describe 'Symnet', which achieves invariance to symmetry group actions. Equivariance to discrete group actions was achieved through parameter sharing in Ravanbakhsh et al. (2017). For the case of data on a spherical domain, one considers exploiting equivariance to the rotation

group. Several research groups recently reported spherical-CNNs (SCNNs) to accommodate such an equivariance in defining the convolution of functions (Worrall et al., 2017; Cohen & Welling, 2016; Cohen et al., 2018). In another recent work Esteves et al. (2017), authors describe a polar transformer network, which is equivariant to rotations and scaling transformations. By combining this with a spatial transformer (Jaderberg et al., 2015), they achieve equivariance to translations as well. More generally, equivariance of convolution operations to group actions admitted by the underlying manifold is what is needed to this end and most recent work reported in Chakraborty et al. (2018); Kondor & Trivedi (2018) achieves this for Riemannian homogeneous spaces.

In this paper we will consider the second problem, namely, when the input data are sample points on known Riemannian manifolds for example, the manifold of symmetric positive definite matrices, $SPD(n)$, the special orthogonal group, $\mathsf{SO}(n)$, the $n$-sphere, $\mathbf{S}^n$, the Grassmannian, $\mathsf{Gr}(p, n)$, and others. To be precise, the domain of interest is an $n$-dimensional field of points sampled from a Riemannian manifold. There is very little prior work that we are aware of on DNNs that can cope with input data samples residing on these manifolds with the exception of Huang et al. (2016); Huang & Van Gool (2017). In Huang et al. (2016), authors presented a deep network architecture for classification of hand-crafted features residing on a Grassmann manifold that form the input to the network. In Huang & Van Gool (2017), authors presented a deep network architecture for data on $SPD(n)$. In both of these works, the architecture does not involve the use of any convolution or equivalent operations on $\mathsf{Gr}(p, n)$ or $SPD(n)$. Further, it does not use the natural invariant metric or intrinsic operations on the Grassmannian or the $SPD(n)$ in the network blocks. Using intrinsic operations within the layers guarantees that the result remains on the manifold and hence does not require any projection (extrinsic) operations to ensure the result lies in the same space. Further, using extrinsic operations can yield results that are susceptible to significant inaccuracies when the data variance is large (Salehian et al., 2015). Moreover, since there are no convolution type operations defined for data on these manifolds in their network, it can not be considered a generalization to the CNN and as a consequence does not consider equivariance property to the action of the group of isometries denoted by $I(\mathcal{M})$, admitted by the manifold $\mathcal{M}$.

There are several deep networks reported in literature to deal with cases when data reside on 2-manifolds encountered in Computer Vision and Graphics for modeling shapes of objects. Some of these are based on graph-based representations of points on the surfaces in 3D and a generalization of CNNs to graphs (Henaff et al., 2015; Defferrard et al., 2016). There is also recent work in Masci et al. (2015) where the authors presented a deep network called geodesic CNN (GCNN), where convolutions are performed in local geodesic polar charts constructed on the manifold. For more literature on deep networks for data on 2-manifolds, we refer the interested reader to a recent survey paper Bronstein et al. (2017) and references therein.

In this paper, we present a novel DNN framework called the ManifoldNet. This is a potential analog of a CNN and can cope with input data sampled from a Riemannian manifold. The intuition in defining the analog relies on the equivariance property. Note that convolution of functions in vector spaces are equivariant to translations in the spatial domain and in the pixel domain. I.e. if the input pixels are all translated by a fixed amount, then the output pixels are equally translated. Further, it is easy to show that traditional convolutions are equivalent to computing the weighted mean (Goh et al., 2011). Hence, for the case of manifold-valued data, we can define the analogous operation of a weighted Fréchet mean (wFM) and prove that it is equivariant to the action of $I(\mathcal{M})$. This will be achieved in a subsequent section. Our key contributions in this work are: [**presented in section 2**](i) we define the analog of convolution operations for manifold-valued data to be one of estimating the wFM for which we present a provably convergent, efficient and recursive estimator. (ii) A proof of equivariance of wFM to the action of $I(\mathcal{M})$. This equivariance allows the network to share weights within the layers. (iii) A novel deep architecture involving the Riemannian counterparts to the conventional CNN units. [**presented in section 3**] (iv) Several real data experiments on classification and reconstruction demonstrating the performance of the ManifoldNet.

## 2    GROUP ACTION EQUIVARIANT NETWORK FOR MANIFOLD-VALUED DATA

In this section, we will define the equivalent of a convolution operation on Riemannian manifolds. As mentioned in the introduction, the domain of interest is an $n$-dimensional field of manifold valued points. Before formally defining such an operation and building the DNN for the manifold-valued

data, dubbed a ManifoldNet, we first present some relevant concepts from differential geometry that will be used in the rest of the paper.

**Preliminaries.** Let $(\mathcal{M}, g^{\mathcal{M}})$ be a orientable complete Riemannian manifold with a Riemannian metric $g^{\mathcal{M}}$, i.e., $(\forall x \in \mathcal{M})$ $g_x^{\mathcal{M}} : T_x\mathcal{M} \times T_x\mathcal{M} \to \mathbf{R}$ is a bi-linear symmetric positive definite map, where $T_x\mathcal{M}$ is the tangent space of $\mathcal{M}$ at $x \in \mathcal{M}$. Let $d : \mathcal{M} \times \mathcal{M} \to [0, \infty)$ be the metric (distance) induced by the Riemannian metric $g^{\mathcal{M}}$. With a slight abuse of notation we will denote a Riemannian manifold $(\mathcal{M}, g^{\mathcal{M}})$ by $\mathcal{M}$ unless specified otherwise. Let $\Delta$ be the supremum of the sectional curvatures of $\mathcal{M}$.

**Definition 1.** *Let $p \in \mathcal{M}$, $r > 0$. Define $\mathcal{B}_r(p) = \{q \in \mathcal{M} | d(p, q) < r\}$ to be a open ball at $p$ of radius $r$.*

**Definition 2.** *(Groisser, 2004) The local injectivity radius at $p \in \mathcal{M}$, $r_{inj}(p)$, is defined as $r_{inj}(p) = \sup \{r | Exp_p : (\mathcal{B}_r(\mathbf{0}) \subset T_p\mathcal{M}) \to \mathcal{M}$ is defined and is a diffeomorphism onto its image$\}$. The injectivity radius Manton (2004) of $\mathcal{M}$ is defined as $r_{inj}(\mathcal{M}) = \inf_{p \in \mathcal{M}} \{r_{inj}(p)\}$.*

Within $\mathcal{B}_r(p)$, where $r \leq r_{\text{inj}}(\mathcal{M})$, the mapping $\text{Exp}_p^{-1} : \mathcal{B}_r(p) \to \mathcal{U} \subset T_p\mathcal{M}$, is called the inverse Exponential/ Log map.

**Definition 3.** *(Kendall, 1990) An open ball $\mathcal{B}_r(p)$ is a regular geodesic ball if $r < r_{inj}(p)$ and $r < \pi / \left( 2\Delta^{1/2} \right)$.*

In Definition 3 and below, we interpret $1/\Delta^{1/2}$ as $\infty$ if $\Delta \leq 0$. It is well known that, if $p$ and $q$ are two points in a regular geodesic ball $\mathcal{B}_r(p)$, then they are joined by a unique geodesic within $\mathcal{B}_r(p)$ (Kendall, 1990).

**Definition 4.** *(Chavel, 2006) $\mathcal{U} \subset \mathcal{M}$ is strongly convex if for all $p, q \in \mathcal{U}$, there exists a unique length minimizing geodesic segment between $p$ and $q$ and the geodesic segment lies entirely in $\mathcal{U}$.*

**Definition 5.** *(Groisser, 2004) Let $p \in \mathcal{M}$. The local convexity radius at $p$, $r_{cvx}(p)$, is defined as $r_{cvx}(p) = \sup \{r \leq r_{inj}(p) | \mathcal{B}_r(p)$ is strongly convex$\}$. The convexity radius of $\mathcal{M}$ is defined as $r_{cvx}(\mathcal{M}) = \inf_{p \in \mathcal{M}} \{r_{cvx}(p)\}$.*

For the rest of the paper, we will assume that the samples on $\mathcal{M}$ lie inside an open ball $U = \mathcal{B}_r(p)$ where $r = \min \{r_{\text{cvx}}(\mathcal{M}), r_{\text{inj}}(\mathcal{M})\}$, for some $p \in \mathcal{M}$, unless mentioned otherwise. Now, we are ready to define the operations necessary to develop the ManifoldNet.

## 2.1 wFM ON $\mathcal{M}$ AS A GENERALIZATION OF CONVOLUTION

We will now define a convolution type operation on points sampled from $\mathcal{M}$. This convolution operation will perform an averaging operation over a moving window, but will replace weighted sums with weighted intrinsic averages. Let $\{X_i\}_{i=1}^N$ be the manifold-valued samples on $\mathcal{M}$. We define the convolution type operation on $\mathcal{M}$ as the weighted Fréchet mean (wFM) (Maurice Fréchet, 1948) of the samples $\{X_i\}_{i=1}^N$. Also, by the aforementioned condition on the samples, the existence and uniqueness of FM is guaranteed (Afsari, 2011). As mentioned earlier, it is easy to show (see Goh et al. (2011)). that convolution $\psi^* = b \star a$ of two functions $a : X \subset \mathbf{R}^n \to \mathbf{R}$ and $b : X \subset \mathbf{R}^n \to \mathbf{R}$ can be formulated as computation of the weighted mean $\psi^* = argmin_\psi \int a(\mathbf{u})(\psi - \widetilde{b}_\mathbf{u})^2 d\mathbf{u}$, where, $\forall \mathbf{x} \in X, \widetilde{b}_\mathbf{u}(\mathbf{x}) = b(\mathbf{u} + \mathbf{x})$ and $\int a(\mathbf{x})d\mathbf{x} = 1$. Here, $f^2$ for any function $f$ is defined pointwise. Further, the defining property of convolutions in vector spaces is the linear translation equivariance. Since weighted mean in vector spaces can be generalized to wFM on manifolds and further, wFM can be shown (see below) to be equivariant to group actions admitted by the manifold, we claim that wFM is a generalization of convolution operations to manifold-valued data.

Let $\{w_i\}_{i=1}^N$ be the weights such that they satisfy the convexity constraint, i.e., $\forall i, w_i > 0$ and $\sum_i w_i = 1$, then wFM, $\textsf{wFM}(\{X_i\}, \{w_i\})$ is defined as:

$$\textsf{wFM}(\{X_i\}, \{w_i\}) = \operatorname*{argmin}_{M \in \mathcal{M}} \sum_{i=1}^N w_i d^2(X_i, M) \tag{1}$$

Analogous to the equivariance property of convolution translations in vector spaces, we will now proceed to show that the wFM is equivariant under the action of the group of isometries of $\mathcal{M}$. We

will first formally define the group of isometries of $\mathcal{M}$ (let us denote it by $G$) and then define the equivariance property and show that wFM is $G$-equivariant.

**Definition 6** (**Group of isometries of** $\mathcal{M}$ ($I(\mathcal{M})$)). *A diffeomorphism $\phi : \mathcal{M} \to \mathcal{M}$ is an isometry if it preserves distance, i.e., $d(\phi(x), \phi(y)) = d(x, y)$. The set $I(\mathcal{M})$ of all isometries of $\mathcal{M}$ forms a group with respect to function composition. Rather than write an isometry as a function $\phi$, we will write it as a group action. Henceforth, let $G$ denote the group $I(\mathcal{M})$, and for $g \in G$, and $x \in \mathcal{M}$, let $g.x$ denote the result of applying the isometry $g$ to point $x$.*

Clearly $\mathcal{M}$ is a $G$ set (see Dummit & Foote (2004) for the definition of a $G$ set). We will now define equivariance and show that wFM, is $G$-equivariant.

**Definition 7** (**Equivariance**). *Let $X$ and $Y$ be $G$ sets. Then, $F : X \to Y$ is said to be $G$-equivariant if $\forall g \in G$, $\forall x \in X$, $F(g.x) = g.F(x)$.*

Let $U \subset \mathcal{M}$ be an open ball inside which FM exists and is unique, let $P$ be the set consists of all possible finite subsets of $U$.

**Theorem 1.** *Given $\{w_i\}$ satisfying the convex constraint, let $F : P \to U$ be a function defined by $\{X_i\} \mapsto \textsf{wFM}(\{X_i\}, \{w_i\})$. Then, $F$ is $G$-equivariant.*

*Proof.* Let $g \in G$ and $\{X_i\}_{i=1}^N \in P$, now, let $M^* = \textsf{wFM}(\{X_i\}, \{w_i\})$, as $g.F(\{X_i\}) = g.M^*$, it suffices to show $g.M^*$ is $\textsf{wFM}(\{g.X_i\}, \{w_i\})$ (assuming the existence and uniqueness of $\textsf{wFM}(\{g.X_i\}, \{w_i\})$ which is stated in the following claim).
**Claim:** Let $U = \mathcal{B}_r(p)$ for some $r > 0$ and $p \in \mathcal{M}$. Then, $\{g.X_i\} \subset \mathcal{B}_r(g.p)$ and hence $\textsf{wFM}(\{g.X_i\}, \{w_i\})$ exists and is unique.

Let $\widetilde{M}$ be $\textsf{wFM}(\{g.X_i\}, \{w_i\})$. Then, $\sum_{i=1}^N w_i d^2\left(g.X_i, \widetilde{M}\right) = \sum_{i=1}^N w_i d^2\left(X_i, g^{-1}.\widetilde{M}\right)$. Since, $M^* = \textsf{wFM}(\{X_i\}, \{w_i\})$, hence, $M^* = g^{-1}.\widetilde{M}$, i.e., $\widetilde{M} = g.M^*$. Thus, $g.M^* = \textsf{wFM}(\{g.X_i\}, \{w_i\})$, which implies $F$ is $G$-equivariant. ∎

Now we give some examples of $\mathcal{M}$ with the corresponding group of isometries $G$. Let $\mathcal{M} = \textsf{SPD}(n)$ (the space of $n \times n$ symmetric positive-definite matrices). Let $d$ be the Stein metric on $\textsf{SPD}(n)$. Then, the group of isometries $G$ is $\textsf{O}(n)$ (the space of $n \times n$ orthogonal matrices). A class of Riemannian manifolds on which $G$ acts transitively are called Riemannian homogeneous spaces. We can see that on a Riemannian homogeneous space $\mathcal{M}$, wFM is $G$-equivariant. Equipped with a $G$-equivariant operator on $\mathcal{M}$, we can claim that the wFM (defined above) is a valid convolution operator since group equivariance is a unique defining property of a convolution operator. The rest of this subsection will be devoted to developing an efficient way to compute wFM. Let $\omega^{\mathcal{M}} > 0$ be the Riemannian volume form. Let $p_{\mathbf{X}}$ be the probability density of a $U$-valued random variable $\mathbf{X}$ with respect to $\omega^{\mathcal{M}}$ on $U \subset \mathcal{M}$, so that $\textsf{Pr}(X \in \mathfrak{A}) = \int_{\mathfrak{A}} p_X(Y)\omega^{\mathcal{M}}(Y)$ for any Borel-measurable subset $\mathfrak{A}$ of U. Let $Y \in U$, we can define the expectation of the real valued random variable $d^2(, Y) : U \to \mathbf{R}$ by $E\left[d^2(, Y)\right] = \int_U d^2(X, Y)p_{\mathbf{X}}(X)\omega^{\mathcal{M}}(X)$. Now, let $w : U \to (0, \infty)$ be an integrable function and $\int_U w(X)\omega^{\mathcal{M}}(X) = 1$.

Then, observe that, $E_w\left[d^2(, Y)\right] := \int_U w(X)d^2(X, Y)p_X(X)\omega^{\mathcal{M}}(X) = C \int_U d^2(X, Y)\widetilde{p}_X(X) \omega^{\mathcal{M}}(X) = C \widetilde{E}\left[d^2(, Y)\right]$. Here, $\widetilde{p}_X$ is the probability density corresponding to the probability measure $\widetilde{\textsf{Pr}}$ defined by, $\widetilde{\textsf{Pr}}(X \in \mathfrak{X}) = \int_{\mathfrak{X}} \widetilde{p}_X(Y)\omega^{\mathcal{M}}(Y) := \int_{\mathfrak{X}} \frac{1}{C} p_X(Y)w(Y)\omega^{\mathcal{M}}(Y)$, where, $\mathfrak{X}$ lies in the Borel $\sigma$-algebra over $U$ and $C = \int_U p_X(Y)w(Y)\omega^{\mathcal{M}}(Y)$. Note that the constant $C > 0$, since $p_X$ is a probability density, $w > 0$ and $\mathcal{M}$ is orientable. Thus, $E_w\left[d^2(, Y)\right]$ with respect to $p_X$ is proportional to $\widetilde{E}\left[d^2(, Y)\right]$ with respect to $\widetilde{p}_X$.

Now, we will state the following proposition (the proof is in the appendix section).

**Proposition 2.** *(i) $\textsf{supp}(p_X) = \textsf{supp}(\widetilde{p}_X)$. (ii) $\textsf{wFE}(X, w) = \textsf{FE}\left(\widetilde{X}\right)$.*

Let $\{X_i\}_{i=1}^N$ be samples drawn from $p_X$ and $\left\{\widetilde{X}_i\right\}_{i=1}^N$ be samples drawn from $\widetilde{p}_X$. In order to compute wFM, we will now present an online algorithm (inductive FM Estimator – dubbed iFME).

Given, $\{X_i\}_{i=1}^N \subset U$ and $\{w_i := w(X_i)\}_{i=1}^N$ such that $\forall i, w_i > 0$, the $n^{th}$ estimate, $M_n$ of wFM $(\{X_i\}, \{w_i\})$ is given by the following recursion:

$$M_1 = X_1 \qquad\qquad M_n = \Gamma_{M_{n-1}}^{X_n}\left(\frac{w_n}{\sum_{j=1}^n w_j}\right). \qquad (2)$$

In the above equation, $\Gamma_X^Y : [0,1] \to U$ is the shortest geodesic curve from $X$ to $Y$. Observe that, in general wFM is defined with $\sum_{i=1}^N w_i = 1$, but in above definition, $\sum_{i=1}^N w_i \neq 1$. We can normalize $\{w_i\}$ to get $\{\widetilde{w}_i\}$ by $\widetilde{w}_i = w_i/(\sum_i w_i)$, but then Eq. 2 will not change as $\widetilde{w}_n/\left(\sum_{j=1}^n \widetilde{w}_j\right) = w_n/\left(\sum_{j=1}^n w_j\right)$. This gives us an efficient inductive/recursive way to define convolution operation on $\mathcal{M}$. Now, we state that the proposed wFM estimator is consistent (the proof is in the appendix).

**Proposition 3.** *Using the above notations and assumptions, let $\{X_i\}_{i=1}^N$ be i.i.d. samples drawn from $p_X$ on $\mathcal{M}$. Let the wFE be finite. Then, $M_N$ converges a.s. to wFE as $N \to \infty$.*

## 2.2 Nonlinear operation between wFM-layers for $\mathcal{M}$-valued Data

In the traditional CNN model, we need a nonlinear function between two convolutional layers similar to ReLU and softmax. As argued in Mallat (2016), any nonlinear function used in CNN is basically a contraction mapping. Formally, let $F$ be a nonlinear mapping from $U$ to $V$. Let assume, $U$ and $V$ are metric spaces equipped with metric $d_U$ and $d_V$ respectively. Then, $F$ is a contraction mapping *iff* $\exists c < 1$ such that, $d_V(F(x), F(y)) \leq c\, d_U(x, y)$. $F$ is a non-expansive mapping (Mallat, 2016) *iff* $d_V(F(x), F(y)) \leq d_U(x, y)$.

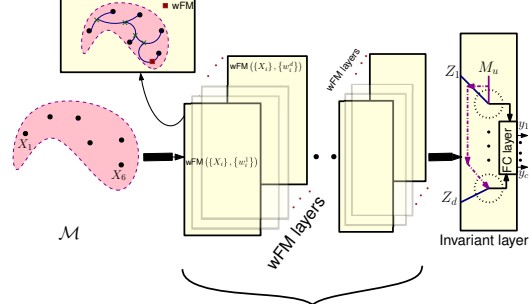

One can easily see that the popular choices for nonlinear operations like ReLU, sigmoid are in fact non-expansive mappings. Now, we will show that the function wFM as defined in 1, is a contraction mapping for non-trivial choices of weights. Let $\{X_i\}_{i=1}^N$ and $\{Y_j\}_{j=1}^M$ be the two set of samples on $\mathcal{M}$. Without any loss of generality, assume $N \leq M$. We consider the set $\mathcal{U}^M = \underbrace{U \times \cdots \times U}_{M \text{ times}}$. Clearly $\{Y_j\}_{j=1}^M \in \mathcal{U}^M$ and we embed $\{X_i\}_{i=1}^N$ in $\mathcal{U}^M$ as follows: we construct $\left\{\widetilde{X}_i\right\}_{i=1}^M$ from $\{X_i\}_{i=1}^N$ by defining $\widetilde{X}_i = X_{(i-1)\bmod N+1}$. Let us denote the embedding by $\iota$. Now, define the distance on $\mathcal{U}^M$ as $d\left(\left\{\widetilde{X}_i\right\}_{i=1}^M, \{Y_j\}_{j=1}^M\right) = \max_{i,j} d(X_i, Y_j)$. The choice of weights for wFM is said to be trivial if one of the weights is 1 and hence the rest are 0.

Figure 1: Schematic diagram of ManifoldNet

**Proposition 4.** *For all nontrivial choices of $\{\alpha_i\}_{i=1}^N$ and $\{\beta_j\}_{j=1}^M$ satisfying the convexity constraint , $\exists c < 1$ such that,*

$$d\left(wFM\left(\{X_i\}_{i=1}^N, \{\alpha_i\}_{i=1}^N\right), wFM\left(\{Y_j\}_{i=1}^M, \{\beta_j\}_{i=1}^M\right)\right) \leq c\, d\left(\iota\left(\{X_i\}_{i=1}^N\right), \{Y_j\}_{j=1}^M\right) \quad (3)$$

## 2.3 The invariant (last) layer

We will form a deep network by cascading multiple wFM blocks each of which acts as a convolution-type layer. Each convolutional-type layer is equivariant to the group action, and hence at the end of the cascaded convolutional layers, the output is equivariant to the group action applied to the input of the network. Let $d$ be the number of output channels each of which outputs a wFM, hence each of the channels is equivariant to the group action. However, in order to build a network that yields an output which is invariant to the group action, we now seek the last layer (i.e., the analogue to a linear classifier) to be invariant to the group action. The last layer is thus constructed as follows: Let $\{Z_1, \cdots, Z_d\} \subset \mathcal{M}$ be the output of $d$ channels and $M_u = \mathsf{FM}\left(\{Z_i\}_{i=1}^d\right) =$

$\mathsf{wFM}\left(\{Z_i\}_{i=1}^d, \{1/d\}_1^d\right)$ be the unweighted FM of the outputs $\{Z_i\}_{i=1}^d$. Then, we construct a layer with $d$ outputs whose $i^{th}$ output $o_i = d(M_u, Z_i)$. Let $c$ be the number of classes for the classification task, then, a fully connected (FC) layer with inputs $\{o_i\}$ and $c$ output nodes is build. A softmax operation is then used at the $c$ output nodes to obtain the outputs $\{y_i\}_{i=1}^c$. In the following proposition we claim that this last layer with $\{Z_i\}_{i=1}^d$ inputs and $\{y_i\}_{i=1}^c$ outputs is group invariant.

**Proposition 5.** *The last layer with $\{Z_i\}_{i=1}^d$ inputs and $\{y_i\}_{i=1}^c$ outputs is group invariant.*

In Fig. 1 we present a schematic of ManifoldNet depicting the different layers of processing the manifold-valued data as described above in Sections 2.1-2.3.

## 3 EXPERIMENTS

In this section we present performance of the ManifoldNet framework on several standard computer vision problems. The breadth of application coverage here includes classification and reconstruction problems. We begin with a video classification problem and then present a reconstruction problem using an auto-encoder-decoder set up.

### 3.1 VIDEO CLASSIFICATION

We start by using the method in Yu & Salzmann (2017) which we summarize here. Given a video with dimensions $F \times 3 \times H \times W$ of $F$ frames, 3 color channels and a frame size of $H \times W$, we can apply a convolution layer to obtain an output of size $F \times C \times H' \times W'$ consisting of $C$ channels of size $H' \times W'$. We compute the covariance matrix of the channels to obtain a sequence of $F$ symmetric positive (semi) definite matrices of size $C \times C$.

From here we can apply a series of temporal ManifoldNet wFMs to transform the $F \times C \times C$ input to a temporally shorter $F' \times K \times C \times C$ output, where $K$ is the number of wFM channels. Within the temporal ManifoldNet wFMs we use a simple weight normalization to ensure that the weights are within $[0, 1]$, and for the weights $w_i$ of any output channel we add a weight penalty of the form $(\sum w_i - 1)^2$ to the loss function to ensure that we obtain a proper wFM. We then reshape this to $F'K \times C \times C$ and pass it through an invariant final layer (section 2.3) to obtain a vector of size $F'K$. Finally, a single FC+SoftMax layer is applied to produce a classified output. We call this the SPD temporal convolutional architecture network (SPD-TCN). Figure 2 illustrates the network architecture described above. In general, the SPD-TCN tends to perform very well on video classification tasks while *using very few parameters*, and runs efficiently due to the wFM structure.

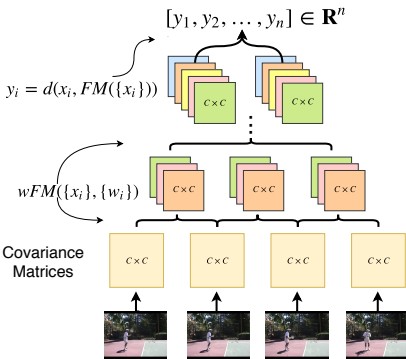

Figure 2: SPD-TCN Network Architecture

We tested the SPD-TCN on the Moving MNIST dataset (Srivastava et al., 2015). Recently, in Chakraborty et al. (2018) authors developed a manifold valued recurrent network architecture, dubbed SPD-SRU, which produced state-of-the-art classification results on Moving MNIST dataset in comparison to LSTM (Hochreiter & Schmidhuber, 1997), SRU (Oliva et al., 2017), TT-LSTM and TT-GRU (Yang et al., 2017) networks. For the LSTM and SRU networks, convolution layers are also used before the recurrent unit. We will compare directly with the results presented in Chakraborty et al. (2018). For details of the various architectures used please see section 5 of Chakraborty et al. (2018). We will also compare with Huang & Van Gool (2017), which proposes a set of layers for learning SPD matrices, dubbed SPDNet. When using SPDNet we first downsample

| Mode | # params. | time (s) / epoch | orientation (°) | | |
|---|---|---|---|---|---|
| | | | 30-60 | 10-15 | 10-15-20 |
| SPD-TCN | **738** | $\sim 2.7$ | **1.00 ± 0.00** | **0.99 ± 0.01** | **0.97 ± 0.02** |
| SPD-SRU | 1559 | $\sim 6.2$ | **1.00 ± 0.00** | 0.96 ± 0.02 | 0.94 ± 0.02 |
| TT-GRU | 2240 | $\sim 2.0$ | **1.00 ± 0.00** | 0.52 ± 0.04 | 0.47 ± 0.03 |
| TT-LSTM | 2304 | $\sim 2.0$ | **1.00 ± 0.00** | 0.51 ± 0.04 | 0.37 ± 0.02 |
| SRU | 159862 | $\sim 3.5$ | **1.00 ± 0.00** | 0.75 ± 0.19 | 0.73 ± 0.14 |
| SPDNet | 110000 | $\sim 30.2$ | 1.00 ± 0.03 | 0.49 ± 0.02 | 0.39 ± 0.01 |
| LSTM | 252342 | $\sim 4.5$ | 0.97 ± 0.01 | 0.71 ± 0.07 | 0.57 ± 0.13 |

Table 1: Comparison results on Moving MNIST

the videos to $20 \times 20$ frames and compute the covariance matrix of the frames, which is then fed to an architecture similar to that of the emotion recognition experiment of Huang & Van Gool (2017) Specifically, we use three blocks of BiMap/ReEig layers, since such a configuration gave the best validation performance. The transformation sizes for these blocks are set to $400 \times 200$, $200 \times 100$ and $100 \times 50$ respectively. The Moving MNIST data generated in Srivastava et al. (2015) consists of 1000 samples, each of 20 frames. Each sample shows two randomly chosen MNIST digits moving within a $64 \times 64$ frame, with the direction and speed of movement fixed across all samples in a class.

The speed is kept the same across different classes, but the digit orientation will differ across two different classes. We summarize the 10-fold cross validation results for several orientation differences between classes in Table 1. For this experiment the SPD-TCN will consist of a single wFM layer with kernel size 5 and stride 3 returning 8 channels, making for an $8 \times 8$ covariance matrix. We then apply three temporal SPD wFM layers of kernel size 3 and stride 2, with the following channels $1 \to 4 \to 8 \to 16$, i.e. after these three temporal SPD wFMs we have 16 temporal channels. This $16 \times 8 \times 8$ is used as an input to the invariant final layer to get a 16 dimensional output vector, which is transformed by a FC+SoftMax layer to obtain the output.

## 3.2 DIMENSIONALITY REDUCTION

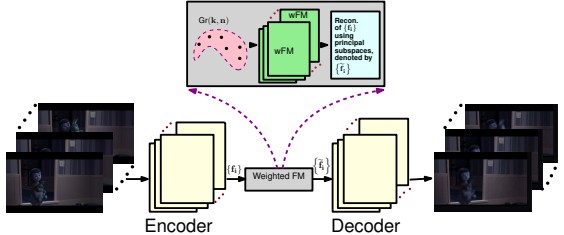

Figure 3: Pictorial description of autoencoder+iFME

Here we present experiments demonstrating the applicability of the theory layed out in Section 2 to the case of linear dimensionality reduction, specifically principal component analysis (PCA), which is the workhorse of many machine learning algorithms. In Chakraborty et al. (2017), authors presented an online subspace averaging algorithm for construction of principal components via intrinsic averaging on the Grassmannian. In this section, we achieve the intrinsic Grassmann averaging process in the framework of ManifoldNet to compute the principal subspaces and achieve the dimensionality reduction. In the context of DNNs, dimensionality reduction is commonly achieved via an autoencoder architecture. More recently, DNNs have shown promising results when the data manifold is intrinsically non-linear, as in the case of natural images. In the deep learning community this has become a field in its own right, known as representation learning or feature learning (Bengio et al., 2013) with works including Vincent et al. (2010), Kingma & Welling (2013) Oord et al. (2016), Van Den Oord et al. (2016)), Kingma & Dhariwal (2018). Many of these architectures are modifications of the traditional autoencoder network, which involves learning an identity map through a small latent space. In our application, we modify the traditional autoencoder model by adding a ManifoldNet layer to perform a learned linear dimensionality reduction in the latent space, although in principal, our techniques can be applied to most autoencoder based models such as the variational autoencoders. To compute a linear subspace in the ManifoldNet framework we use an intrinsic averaging scheme on the Grassmannian. A point on the Grassmannian $\mathsf{Gr}(k, n)$ corresponds to $k$-dimensional subspace of $\mathbf{R}^n$ and thus can be specified by an orthonormal basis $X$. Chakraborty et al. (2017) proposed an efficient intrinsic averaging scheme on $\mathsf{Gr}(k, n)$ that converges to the $k$-dimensional principal subspace of a normally distributed dataset in $\mathbf{R}^n$. In the ManifoldNet framework we can modify this technique to learn a wFM of points on the Grassmannian that corresponds to a subspace of the latent space which minimizes the reconstruction error by using a Grassmannian averaging layer that learns the weights in the wFM. This essentially will give us a lower dimensional representation of the samples after projecting them on to the learned subspace. Note that combining the convergence proof in Chakraborty et al. (2017) and Proposition 2 (ii), we claim that the wFM learned using the ManifoldNet asymptotically converges to the principal subspace. Now, we give a detailed description of our experimental setup to show the applicability of ManifoldNet to dimensionality reduction.

### 3.2.1 VIDEO RECONSTRUCTION EXPERIMENT

A traditional convolutional autoencoder performs non-linear dimensionality reduction by learning an identity function through a small latent space. A common technique used when the desired latent space is smaller than the output of the encoder is to apply a fully connected layer to match dimensions. We replace this fully connected layer by a weighted subspace averaging and projection block, called the Grassmann averaging layer. Specifically, we compute the wFM of the output of the encoder to

get a subspace in the encoder output space. We then project the encoder output onto this space to obtain a reduced dimensionality latent space. In general this offers a significant parameter reduction while also increasing the reconstruction error performance of the autoencoder and giving realistic reconstructions. We call an autoencoder with the Grassmann averaging block

an autoencoder+iFME network, as shown in Fig. 3. In the experiments we compare this to other dimensionality reduction techniques, including regular autoencoders that use fully connected layers to match encoder and latent space dimensions. We begin by testing on a 1000 frame color sample of video from the 1964 film "Santa Clause Conquers the Martians" of frame size $320 \times 240$. Here we use an 8 layer encoding-decoding architecture with **Conv** $\rightarrow$ **ELU** $\rightarrow$ **Batchnorm** layers, with the final layer applying a sigmoid activation to normalize pixel values. The encoder returns a feature video consisting of 128 channels of size 120 for a dimension of $1000 \times 15360$. We compare a fully connected layer to a Grassmann averaging layer, both mapping to a desired latent space of dimension $1000 \times 20$. The per pixel average reconstruction error for the Grassmann block network is 0.0110, compared to 0.0122 for the fully connected network,

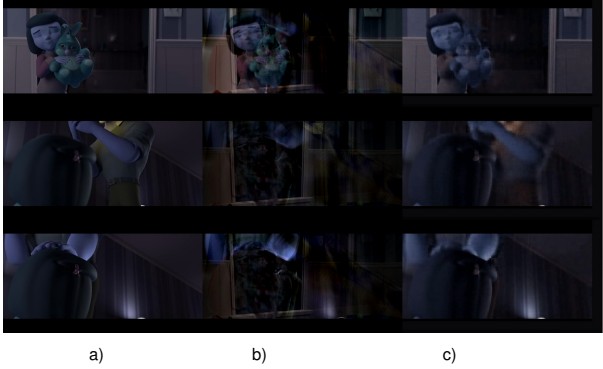

Figure 4: Reconstruction of select movie frames (a) original frame (b) using PCA (c) using iFME+autoencoder

representing an improvement of 10.9%. We also observe a parameter reduction of 46%, which can be attributed to the number of parameters in the large fully connected layer. In Fig. 5, the computation time is plotted against error tolerance for the autoencoder and the iFME+autoencoder. We can see that iFME+autoencoder achieves faster convergence than autoencoder. It is possible to obtain a low reconstruction error on autoencoding tasks and still observe low visual quality reconstructions. To ensure this is not the case we run the same experiment on 300 frames of the $1280 \times 720$ short film [1], with a latent space frame dimension of 300x50. In Fig. 4 we compare the visual quality of our autoencoder to that

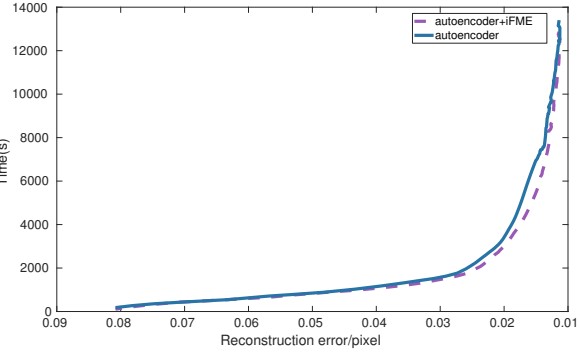

Figure 5: Computation time vs. error tolerance plot comparison between autoencoder and iFME+autoencoder

of PCA with 50 princial components, i.e., we reduce the dimension from $1280 \times 720 \times 3$ to 50. The entire sample reconstruction is shown in [2] (in the same order as in Fig. 4).

## 4 CONCLUSIONS

In this paper, we presented a novel deep network called ManifoldNet suited for processing manifold-valued data sets. Inputs to the ManifoldNet are manifold-valued and not real or complex-valued functions defined on non-Euclidean domains. Our key contributions are: (i) A novel deep network to be perceived as a generalization of the CNN to the case when the input data are manifold-valued using purely intrinsic operations on the manifold where the data reside. (ii) Analogous to convolutions in vector spaces – which can be computed using the weighted mean – we present wFM operations on the manifold and prove the equivariance of the wFM to natural group operations admitted by the manifold. This equivariance allows us to share the learned weights within a layer of the ManifoldNet. (iii) An efficient recursive wFM estimator that is provably convergent is presented. (iv) Experimental results demonstrating the efficacy of the ManifoldNet for, (a) video classification and (b) principal component computation from videos and reconstruction are also presented.

---

[1]https://www.youtube.com/watch?v=t1hMBnIMt5I

[2]https://streamable.com/3yqrx

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

## 5 APPENDIX

**Proposition 2.** *(i)* $\text{supp}\,(p_X) = \text{supp}\,(\widetilde{p}_X)$. *(ii)* $\text{wFE}\,(\mathsf{X}, w) = \text{FE}\,\left(\widetilde{\mathsf{X}}\right)$.

*Proof.* Let $X \in \text{supp}\,(p_X)$, then, $p_X\,(X) > 0$. Since, $w(X) > 0$, hence, $\widetilde{p}_X\,(X) > 0$ and thus, $X \in \text{supp}\,(\widetilde{p}_X)$. On the other hand, assume $\widetilde{X}$ to be a sample drawn from $\widetilde{p}_X$. Then, either $p_X\left(\widetilde{X}\right) = 0$ or $p_X\left(\widetilde{X}\right) > 0$. If, $p_X\left(\widetilde{X}\right) = 0$, then, $\widetilde{p}_X\left(\widetilde{X}\right) = 0$ which contradicts our assumption. Hence, $p_X\left(\widetilde{X}\right) > 0$, i.e., $\widetilde{X} \in \text{supp}\,(p_X)$. This concludes the proof of part (i).

Let $\mathsf{X}$ and $\widetilde{\mathsf{X}}$ be the $\mathcal{M}$ valued random variable following $p_X$ and $\widetilde{p}_X$ respectively. We define the weighted Fréchet expectation (wFE) of $\mathsf{X}$ as:

$$\text{wFE}\,(\mathsf{X}, w) = \underset{Y \in \mathcal{M}}{\text{argmin}} \int_{\mathcal{M}} w(X)d^2(X, Y)p_X(X)\omega^{\mathcal{M}}(X)$$

Observe,

$$\begin{aligned} E_w\left[d^2(, Y)\right] &:= \int_U w(X)d^2(X, Y)p_X(X)\omega^{\mathcal{M}}(X) \\ &= C \int_U d^2(X, Y)\widetilde{p}_X(X)\omega^{\mathcal{M}}(X) \\ &= C\,\widetilde{E}\left[d^2(, Y)\right]. \end{aligned} \tag{4}$$

Hence, we get $\text{FE}\left(\widetilde{\mathsf{X}}\right) = \text{wFE}\,(\mathsf{X}, w)$, as $C$ is independent of the choice of $Y$, which concludes the proof of part (ii). ∎

**Proposition 3.** *Using the notations and assumptions used in the paper, let $\{X_i\}_{i=1}^N$ be i.i.d. samples drawn from $p_X$ on $\mathcal{M}$. Let the wFE be finite. Then, $M_N$ converges a.s. to wFE as $N \to \infty$.*

*Proof.* Using Proposition 2, we know that $\exists\,\widetilde{p}_X$ such that, $\text{wFE}\,(\mathsf{X}, w) = \text{FE}\left(\widetilde{\mathsf{X}}\right)$. Thus, it is enough to show the consistency of our proposed estimator when weights are uniform. In order to prove the consistency, we will split the proof into two cases namely, manifolds with (i) non-positive sectional curvature and (ii) non-negative sectional curvature. The reason for doing this split is so that we can use existing theorems in literature for proving the result. We will use the theorems proved in Sturm (2003) and Bonnabel (2013) for manifolds with non-positive and non-negative sectional curvatures respectively. Note that the proof holds only for manifolds with a uniform sign of sectional curvatures.

**Theorem 6** ($\mathcal{M}$ has non-negative sectional curvature). *Using the above notations, if $\exists A > 0$ such that, $d\,(M_n, X_{n+1}) \leq A$ for all $n$. Then, $M_N$ converges a.s. to wFE as $N \to \infty$ (see Bonnabel (2013) for the proof).*

**Theorem 7** ($\mathcal{M}$ has non-positive sectional curvature). *Using the above notations $M_N$ converges a.s. to wFE as $N \to \infty$ (see Sturm (2003) for the proof).*

∎

**Proposition 5.** *The last layer with $\{Z_i\}_{i=1}^d$ inputs and $\{y_i\}_{i=1}^c$ outputs is group invariant.*

*Proof.* Using the above construction, let $W \in \mathbf{R}^{c \times d}$ and $\mathbf{b} \in \mathbf{R}^c$ be the weight matrix and bias respectively of the FC layer. Then,

$$\begin{aligned} \mathbf{y} &= F\left(W^T\mathbf{o} + \mathbf{b}\right) \\ &= F\left(W^T d\,(M_u, Z) + \mathbf{b}\right), \end{aligned} \tag{5}$$

where, $F$ is the softmax function. In the above equation, we treat $d\,(M_u, Z)$ as the vector $[d\,(M_u, Z_1), \cdots, d\,(M_u, Z_d)]^t$. Observe that, $g.M_u = \text{FM}\left(\{g.Z_i\}_{i=1}^d\right)$. As each of the $d$ channels

is group equivariant, $Z_i$ becomes $g.Z_i$. Because of the property of the distance under group action, $d(g.M_u, g.Z_i) = d(M_u, Z_i)$. Hence, one can see that if we change the inputs $\{Z_i\}$ to $\{g.Z_i\}$, the output $\mathbf{y}$ will remain invariant. ∎

