# OpenReview forum: "MANIFOLDNET: A DEEP NEURAL NETWORK FOR MANIFOLD-VALUED DATA"
_ICLR.cc/2019/Conference_

### Official Review · AnonReviewer1 · 2018-10-31
**Too many things bother me to recommend acceptance**

**Rating:** 4
**Confidence:** 4

**Review:**

Brief summary:
The paper considers a generalization of convolutional neural networks (CNNs) to data residing on Riemannian manifolds. The idea is to replace convolutions with weighted averages, which are implemented intrinsically on the manifold. It is shown that this operator is equivariant to isometric group actions. A related approach for dimensionality reduction is also proposed, but I think this only applies to Euclidean data, so I am a bit confused about that part. Empirical performance is reported on toy data with weak baselines.

Good points about the paper:
+ It is a relevant point that the intrinsic average in manifolds is a way to generalized convolutions to the Riemannian domain.
+ The paper is generally fairly easy to read.

Concerns with the paper:
- A key point of CNNs is that you learn filters that are small, i.e. only have non-zero weight assigned to a few neighboring pixels. As far as I can tell, the here proposed "filters" would be one weight per data point.

- I am concerned about the stacking of multiple "convolution" (wFM) layers: since each layer computes the average of a set of points, then doesn't stacking multiple "convolution" (wFM) layers on top of each other correspond to computing the average of a set of averages? And can this not be computed by a single average? In other words, is a cascade of "convolution" (wFM) layers equivalent to a single layer? Seems like a complicated way of doing shallow learning unless I misunderstand.

- In section 2.2 it is argued that the weighted average (wFM) is a contraction mapping. While I think the proof is correct, I am concerned about the prerequisite assumption that that distance between a set of points X and Y is the *maximal* distance between points in the two sets. Usually, one would define this distance as the *minimal* distance (akin to the Hausdorff distance). It seems that under this more reasonable choice of distance, the proof no longer holds

- Large parts of section 2.1 is devoted to an efficient algorithm for computing weighted averages on manifolds.  Here the text is written such as to indicate that this is a novel contribution of the present paper, even if these results are readily available in the literature. I strongly encourage a re-writing to emphasize that this is a repetition of previous knowledge.

- Proposition 5 does not appear to come with a proof.

- Section 3.2 introduce a new dimensionality reducing layer based on the Grassmann average construction for subspace learning. I was quite confused when reading this. From what I can tell this layer is only applicable when the input data is Euclidean, and as such appears to be unrelated to the rest of the paper.

- At times the paper is rather sloppy written, e.g. fonts are way too small in figures, the dot(.) notation is not defined (e.g. in def. 7), and the citation style is very difficult to read (please use \citet and \citep instead of \cite).

Other comments:
*) The assumption (page 3) that the data reside within a geodesic ball of a certain radius seem quite strong. It would be good to comment on this in the paper.

*) It is not clear to me that the weighted average is a particularly good way to generalize convolution. Yes, I agree, it is *one
 way to generalize, but why should I pick this one in particular?

*) In the experiment in sec. 3.1 the manifold comes from a particular way to extract features from data. In a deep learning context, we would usually learn such features instead of manually designing them. This seem like a general issue, that most often manifold come into play through a manual feature-design, which seem to be at odds with the end-to-end learning mindset. It might be good to have examples where this is not the case.

---

> ### Author Response · Authors · 2018-11-14
> **Thank the reviewer+answers to the comments**
>
> a) A key point of CNNs is that you learn filters that are small ....
>
> Ans: For a field of manifold valued data, our filter size will be exactly like standard CNN, i.e., one weight for one point inside the filter. So for a 5*5 filter, we have 25 weights to learn for wFM. Now, we share the filter across the entire field (analogous to the standard CNN).
>
> b) I am concerned about the stacking of multiple "convolution" (wFM) layers ...
>
> Ans: This is an interesting point and indeed is true, i.e., two wFM layers can be collapsed into one. But the learning becomes much more computationally expensive if one resorts to collapsing. We illustrate this with a toy example here. Let us consider a 4*4 manifold valued field.  Let us say we used 2*2 wFM filters (with stride 2*2). If we use two such wFM layers to get a single manifold valued point, then the total number of parameters is 4+4=8. Now, instead if we compute the mean using a single layer wFM, the number of parameters increases (doubled) to 4*4=16. This indicates that the learning becomes computationally harder even in this simple toy problem. Though theoretically, the collapsing is a valid thing to do, but in practice, it makes the learning of the filter computationally intractable.
>
>
> c) In section 2.2 it is argued that the weighted average (wFM) is a contraction mapping. ...
>
> Ans: The definition of contraction mapping depends on the choice of metric on both sides of the inequality. So, this is the reason for the choice of “maximal distance” instead of “Hausdroff distance”. Nonetheless, this particular choice of distance is not an assumption but a choice to prove that the mapping is a contraction mapping.
>
>
> d)Large parts of section 2.1 is devoted to an efficient algorithm for computing weighted averages on manifolds ....
>
> Ans: We will rewrite the part for efficient computation for weighted averaging computation to make it explicit that this part is not a contribution of this paper. One of the key contributions is to show that wFM is FM in a different metric and then show that we can use the existing incremental FM algorithm to compute wFM. We will make this clarification in the paper.
>
>
> e) Proposition 5 does not appear to come with a proof.
>
> Ans: We included the proof in the appendix.
>
> f) Section 3.2 introduces a new dimensionality reducing layer based on the Grassmann average construction ....
>
> Ans: This experiment is motivated by Chakraborty'17. The purpose of this experiment is to demonstrate that one can perform PCA (dimensionality reduction) inside a deep network. In Chakraborty'17, the authors used intrinsic Grassmann averages to compute PCs, so this experiment is an example of using a convolution layer to extract PCs, since we defined a convolution layer on a manifold (Grassmannian in this example) using wFM (Grassmann averages in this example).
>
> g) At times the paper is rather sloppy written, e.g. fonts are way too small in figures....
>
> Ans: We thank the reviewer for pointing out the mistakes, we fixed these minor mistakes in the revision.
>
> h) The assumption (page 3) that the data reside within a geodesic ball of a certain radius seems quite strong....
>
> Ans: This assumption is not a hindrance in many applications that we are aware of and is commonly used by many researchers in existing literature. For example, in the case of a hypersphere, the injectivity radius defining this ball is \pi/2 and this results in a fairly large region for consideration in the upper/lower hemisphere. The restriction is imposed to guarantee the uniqueness of the computed FM/wFM.
>
> i) It is not clear to me that the weighted average is a particularly good way to generalize convolution. Yes, I agree, it is *one way to generalize, but why should I pick this one in particular?
>
> Ans: In standard CNN, correlation operator can be computed by using \sum_i w_i x_i inside a filter’s support, where {x_i} are intensities and {w_i} are filter weights. Now, if {w_i} satisfies convexity constraint \sum_i w_i x_i is the minimizer of the weighted variance, i.e., wFM on Euclidean space. This motivates us to generalize the convolution on the manifold using the minimizer of weighted variance, i.e., wFM. Note that the convexity constraint though is not common in standard CNN, is a crucial constraint to ensure that the output of convolution is on the manifold. Also, this convexity constraint can be thought of as a regularizer on the filter.
>
> j) In the experiment in sec. 3.1 the manifold comes from a particular way to extract features from data ....
>
> Ans: We disagree with this comment. In section 3.1., we learned the features inside the network and used the covariance of the extracted features as the input to our manifoldnet. The covariance of the learned features is a choice we made motivated by the work by Yu & Salzmann (2017). Note that, as we train the network in an end-to-end fashion, the features are not hand crafted but learned using the network.

---

> > ### Comment · AnonReviewer1 · 2018-11-23
> > **Follow-up**
> >
> > "a) A key point of CNNs is that you learn filters that are small ....
> >
> > Ans: For a field of manifold valued data, our filter size will be exactly like standard CNN, i.e., one weight for one point inside the filter. So for a 5*5 filter, we have 25 weights to learn for wFM. Now, we share the filter across the entire field (analogous to the standard CNN)."
> >
> > I do not understand this. Does that mean that you work with images where each pixel is manifold-valued, such as what you would see in DTI?

---

> > > ### Author Response · Authors · 2018-11-26
> > > **Response**
> > >
> > > "``I do not understand this. Does that mean that you work with images where each pixel is manifold-valued, such as what you would see in DTI?’’
> > >
> > > Ans:  In these experiments, we do not have a DTI “type” field. Instead we have a time ordered sequence of SPD matrices, which is a 1D field. For example, in the moving MNIST experiment, from one video we get a sequence of SPD matrices, which is treated as one data sample for the ManifoldNet. It should however be noted that our work is applicable to 2D and 3D SPD matrix-valued fields as well.

---

> > ### Comment · AnonReviewer1 · 2018-11-23
> > **Stacking**
> >
> > "b) I am concerned about the stacking of multiple "convolution" (wFM) layers ...
> >
> > Ans: This is an interesting point and indeed is true, i.e., two wFM layers can be collapsed into one. But the learning becomes much more computationally expensive if one resorts to collapsing. We illustrate this with a toy example here. Let us consider a 4*4 manifold valued field.  Let us say we used 2*2 wFM filters (with stride 2*2). If we use two such wFM layers to get a single manifold valued point, then the total number of parameters is 4+4=8. Now, instead if we compute the mean using a single layer wFM, the number of parameters increases (doubled) to 4*4=16. This indicates that the learning becomes computationally harder even in this simple toy problem. Though theoretically, the collapsing is a valid thing to do, but in practice, it makes the learning of the filter computationally intractable."
> >
> > I agree with this comment, but it misses my point (which perhaps was unclear). In the paper it is argued that the weighted mean also serves the role of a contraction mapping, i.e. that a nonlinear activation is not required. My concern regarding stacking layers is that if this is mathematically equivalent to a single layer, then perhaps you actually should be using a more traditional nonlinear activation. If the network depth is purely a computational trick, that does not aid in solving highly nonlinear problems, then I am not convinced about the point of "going deep".

---

> > > ### Author Response · Authors · 2018-11-26
> > > **Response**
> > >
> > > “My concern regarding stacking layers is that if this is mathematically equivalent to a single layer, then perhaps you actually should be using a more traditional nonlinear activation. “
> > >
> > > Ans: Traditional nonlinear activations are not applicable here since they expect the input to be real-valued. For manifold-valued input, some researchers have applied the traditional nonlinear activation after Log-mapping the manifold-valued data into the tangent space anchored at some point on the manifold. We will call this the tangent-ReLU here in this discussion. The reason a tangent-ReLU is not applicable is that after the application of this kind of non-linearity we have to ensure that we can apply the wFM operation in the next layer (or FM  operation for final invariant layer). In order achieve this, we have to ensure that after applying the non-linearity and mapping it back to the manifold using the Riemannian Exp map, the resultant manifold valued points (on which we are going to perform wFM in the next conv./ inv. layer) are inside a geodesic ball (as mentioned in Section 2 of the manuscript). A tangent-ReLU violates such a constraint.
> > >
> > > "``If the network depth is purely a computational trick, that does not aid in solving highly nonlinear problems, then I am not convinced about the point of "going deep”.
> > >
> > > Ans: Note that, in standard CNN the reason for going deeper is two fold, (1) Use non-linearity in between layers to get a non-linear CNN model (2) Efficient usage of network parameters, i.e., equivalence of a deep network and a shallow network with larger number of parameters. Moreover, the reason we cannot collapse a standard CNN is because of the non-linearity in between.
> > >
> > > In our ManifoldNet framework, note that since we defined convolution using wFM, one can easily show that a convolution operator defined in such a way is non-linear on any non-flat manifold. Hence, the need for a further non-linearity is not apparent. Furthermore, we have shown that wFM is not only a non-linear operator but also a contraction mapping, hence it shares some characteristics with standard ReLU like operators. Since wFM, a.k.a the defined convolution operator is non-linear and satisfies the contraction property, we do not have the need for additional nonlinearity. The only need for going deeper is to make the model parameter estimation efficient which is the reason we gave in our earlier response. This however does not rule out introduction of any further nonlinearity if deemed necessary in the future. One such possible non-linear map: $F: \mathcal{M} \rightarrow \mathcal{M}$ is given as follows: $F(x) = g.x$ for some fixed $g \in G$. It is easy to see that this operator $F$ has the following properties: (1) It is non-linear, (2) it can guarantee that, if ${x}$ is within a geodesic ball, so is ${g.x}$.

---

> > > > ### Comment · AnonReviewer1 · 2018-11-26
> > > > **response**
> > > >
> > > > You write:
> > > >
> > > > "In order achieve this, we have to ensure that after applying the non-linearity and mapping it back to the manifold using the Riemannian Exp map, the resultant manifold valued points (on which we are going to perform wFM in the next conv./ inv. layer) are inside a geodesic ball (as mentioned in Section 2 of the manuscript). A tangent-ReLU violates such a constraint."
> > > >
> > > > but previously you wrote:
> > > >
> > > > "h) The assumption (page 3) that the data reside within a geodesic ball of a certain radius seems quite strong....
> > > >
> > > > Ans: This assumption is not a hindrance in many applications that we are aware of and is commonly used by many researchers in existing literature. For example, in the case of a hypersphere, the injectivity radius defining this ball is \pi/2 and this results in a fairly large region for consideration in the upper/lower hemisphere. The restriction is imposed to guarantee the uniqueness of the computed FM/wFM."
> > > >
> > > > These two statements appear to be in direct conflict with each other as far as I can tell.
> > > >
> > > > So far, I don't see this discussion changes my view of the paper, in that there are too many aspects that make me worry about the paper.

---

> > > > > ### Author Response · Authors · 2018-11-27
> > > > > **Response**
> > > > >
> > > > >
> > > > > It appears that the reviewer misunderstood our earlier response. Our response to "h) The assumption (page 3) that the data reside within a geodesic ball of a certain radius seems quite strong....” was to clarify why the constraint is not very strong and not impractical in practice.  This constraint has to be satisfied in order to guarantee the uniqueness of the computed wFM and we are not the first to use this constraint in literature. All researchers that have used FM or wFM computation from data in published literature have and must use this assumption for the uniqueness of the computed estimates.
> > > > >
> > > > > In our response regarding the inclusion of tangent-ReLU, we tried to explain why after the application of a simple extrinsic non-linearity such as a tangent-ReLU, the geodesic ball constraint is violated. This is primarily because, application of an arbitrary extrinsic nonlinearity to a manifold-valued data point cannot guarantee the result to lie inside the geodesic ball.  We also provided in our earlier response a possible way to achieve this without violating the geodesic ball containment constraint. However, we did not find the need to use it in our experiments since we proved that the wFM is already a nonlinear operation and possesses the required contraction property of a ReLU type unit. This however does not rule out our using an additional nonlinearity if deemed necessary in the future.
> > > > >
> > > > > These two responses do not contradict each other at all. The former is explaining why the assumption is not impractical while the later explains why tangent-ReLU type operation can violate this constraint. We hope the reviewer finds the above elaborated responses to this issue of concern more palatable and request that he/she read it with care.
> > > > >
> > > > > “So far, I don't see this discussion changes my view of the paper, in that there are too many aspects that make me worry about the paper.”
> > > > >
> > > > > Ans: This is very unfortunate, and we request the reviewer to list out for us the “too many aspects” stated above that are yet unanswered to the reviewer’s satisfaction. We would be happy to settle any and all issues of concern regarding this work to this reviewer.

---

> > > > > > ### Author Response · Authors · 2018-12-01
> > > > > > **Proof of non-equivalence of single-layer and multi-layer ManifoldNets**
> > > > > >
> > > > > > Here we present a counterexample to the claim by Reviewer-1 that
> > > > > > our multi-layer ManifoldNet cannot be collapsed to a single layer
> > > > > > ManifoldNet. Although we agreed with the Reviewer in our earlier
> > > > > > response, upon more careful thought about this interesting question,
> > > > > > we found that this is only true for constant curvature manifolds
> > > > > > such as Euclidean space (zero curvature), the hypersphere and the
> > > > > > hyperbolic space. For the Euclidean case, the proof is obvious and we will
> > > > > > not provide the proof for the latter two manifolds here. Instead, we
> > > > > > provide a proof via a counterexample that the claim by Reviewer-1 is untrue
> > > > > > for non-constant curvature manifolds such as SPD(n), which is a
> > > > > > negatively curved Riemannian manifold.
> > > > > >
> > > > > > Let us suppose we are given 4 symmetric positive definite (SPD)
> > > > > > matrices,
> > > > > > $A=
> > > > > > \begin{bmatrix}
> > > > > > 0.9593 & 0.3429 \\
> > > > > > 0.3429 & 0.1493
> > > > > > \end{bmatrix}$,
> > > > > >
> > > > > > $B=
> > > > > > \begin{bmatrix}
> > > > > > 1.2575 & 0.5475 \\
> > > > > > 0.5475 & 1.8143
> > > > > > \end{bmatrix}$,
> > > > > >
> > > > > > $C=
> > > > > > \begin{bmatrix}
> > > > > > 1.2435 & 0.6396 \\
> > > > > > 0.6396 & 1.1966
> > > > > > \end{bmatrix}$,
> > > > > >
> > > > > > $D=
> > > > > > \begin{bmatrix}
> > > > > > 1.2511 & 0.5446 \\
> > > > > > 0.5447 & 1.3517
> > > > > > \end{bmatrix}$,
> > > > > >
> > > > > > whose wFM we want to compute. Let us consider
> > > > > > two sequences $S1 = \left\{A,B,C,D\right\}$ and $S2 =
> > > > > > \left\{A,C,B,D\right\}$.  Consider a one layer ManifoldNet for
> > > > > > computing the wFM of these four matrices. For simplicity of
> > > > > > exposition, suppose this one layer network learns equal weights
> > > > > > $(=0.25)$ for all matrices and hence yields the wFM
> > > > > > $M=
> > > > > > \begin{bmatrix}
> > > > > > 1.1640 & 0.4667 \\
> > > > > > 0.4667 & 0.6388
> > > > > > \end{bmatrix}$
> > > > > > as the solution for both sequences $S1$ and $S2$ respectively. To compute the wFM, we use a gradient descent applied to the weighted sum of square geodesic
> > > > > > distances between the unknown wFM and the sample points.
> > > > > >
> > > > > > Now, let us consider a two layer wFM. For $S1$, the first layer
> > > > > > computes wFM of $\left\{A, B\right\}$ and $\left\{C, D\right\}$
> > > > > > respectively and returns $M1$ and $M2$ as the wFMs. Then, the second
> > > > > > layer takes $M1$ and $M2$ as inputs and returns their wFM say,
> > > > > > $M3$. Analogously for the sequence $S2$, the first layer computes wFM
> > > > > > of $\left\{A, C\right\}$ and $\left\{B, D\right\}$ and returns
> > > > > > $\bar{M1}$ and $\bar{M2}$. Then the second layer takes as input,
> > > > > > $\bar{M1}$ and $\bar{M2}$ and returns $\bar{M3}$ as their wFM.
> > > > > >
> > > > > > It can be verified that for the first layer if we use equal weights,
> > > > > > we need the weights for the second layer to be $0.4980$ and $0.5050$
> > > > > > for $S1$ and $S2$ respectively such that both $M3=M$ and
> > > > > > $\bar{M3}=M$. This counterexample shows that the weights are
> > > > > > dependent on the data matrices, which are points on the SPD
> > > > > > manifold. Thus, for each data set, in order for one to collapse the
> > > > > > multi-layer ManifoldNet to a single layer ManifoldNet, one needs a
> > > > > > distinct set of weights that are dependent on the data. This is unlike
> > > > > > in the Euclidean case, where, the weights interact in a known fixed
> > > > > > way for which there is an obvious analytic expression. In the case of
> > > > > > Riemannian manifolds with non-constant curvature, such as SPD(n) or
> > > > > > the Grassmanian, Stiefel, and others, this is not true.
> > > > > >
> > > > > > This clearly shows that the two-layer ManifoldNet cannot be collapsed
> > > > > > to a single layer.

---

### Official Review · AnonReviewer3 · 2018-11-02
**missing important related works, careless writing and insufficient evaluation**

**Rating:** 4
**Confidence:** 4

**Review:**

This paper introduces a generalization of convolution operations to manifold-valued data using  the computation of the weighted Frechet mean (wFM). Without applying any non-linear units and pooling layers, the paper proposes to merely stack suching generalized convolutional layers to construct a deep network for the data residing on Riemannian manifolds. The evaluations on video classification and reconstruction tasks show the advantages of the introduced network over some baselines.

The paper’s major contribution is using wFM to generalize the traditional convolution for manifold-valued data.  Accordingly, the critical technical contribution is to estimate the wFM. The paper suggests a inductive/recursive way for the wFM approximation. To the best of my knowledge, there already exist some inductive/recursive wFM estimation methods like [1,2]. Unfortunately, the authors seem to overlook them and do not discuss them at all. Accordingly, I think the paper missed some important related works for sufficient study.

[1] Y. Lim and M. Pa ́lfia. Weighted inductive means .LAA, 453:59–83, 2014.
[2] Hesamoddin Salehian et al., An efficient recursive estimator of the Fre ́chet mean on a hypersphere with applications to Medical Image Analysis, Mathematical Foundations of Computational Anatomy. 2015.

Due to the inconsistent fonts, chaotic layout it is really hard for the readers to follow the content of the paper. I feel like the paper seems to be completed in the last minute. This brings another critic problem, it is not easy to reimplement the wFM layers, which is the core contribution of the paper. For instance, the paper claims that they used intrinsic Riemannian metric when using wFM to convolve the manifold-valued data. I guess it is involved in \Gamma_X^Y (Eq.2) which is explained as the shortest curve from X to Y. Anyway it fails to describe what kind of intrinsic metric they used for the specific manifold-valued data like SPD matrices and linear subspace. In addition, is it \Gamma_{M_{n-1}}^{X_n} rather than \Gamma_{M_{n-1}}^{X_M} for Eq.(2)?

Another problem is the evaluations are far from sufficient. For video classification, the paper only uses the moving MNIST, which is not a challenging dataset while there are plenty of large scale video datasets. In addition, the paper is expected to compare SOTA video classification methods. To learn the advantage of the proposed network over some related manifold networks like Huang et al., 2016, Huang & Van Gool 2017, it is necessary to evaluate them in the experiments.  For video reconstruction, using 1000 frame color sample of video is also not sufficient to study the effectiveness of the proposed ManifoldNet. Furthemore, it is also expected to compare more SOTA auto-encoder based reconstruction models like VAE [3], AAE [4]  and WAE [5].

[3] Kingma et al., Auto-encoding variational bayes, 2013
[4] Makhzani et al., Adversarial autoencoders, 2015
[5] Wasserstein auto-encoders, 2017

---

> ### Author Response · Authors · 2018-11-14
> **Thank the reviewer+answers to the comments**
>
> a) The paper’s major contribution is using wFM to generalize the traditional convolution for manifold-valued data. Accordingly, the critical technical contribution is to estimate the wFM. The paper suggests a inductive/recursive way for the wFM approximation. To the best of my knowledge, there already exist some inductive/recursive wFM estimation methods like [1,2]. Unfortunately, the authors seem to overlook them and do not discuss them at all. Accordingly, I think the paper missed some important related works for sufficient study.
>
> Ans: The major technical contribution of this paper is not the wFM estimator but is in showing that wFM on manifolds can be used in place of convolutions for manifold-valued data sets (not samples of functions on a manifold). Further, proving that wFM is equivariant to the group actions admitted by the manifold on which the data reside. Additionally, proving that wFM is equivalent to an unweighted FM in a different metric which then allows us to use existing results to prove consistency of the recursive wFM estimator.   Finally, all this was done for a general Riemanian manifold. The references [1], [2] are focused on the manifold of symmetric positive definite matrices (SPD) and hypersphere respectively and hence not valid for other manifolds such as the Grassman, etc. Thus, in this paper, we did not discuss recursive FM estimators in [1,2]. We will gladly include them and place them in the context of the work presented here, in the revision.
>
> b) Due to the inconsistent fonts, chaotic layout it is really hard for the readers to follow the content of the paper.
>
> Ans: In the figure, we have used smaller fonts which we will revise. But can the reviewer please point out the “chaotic layout” that he/she is referring to as we were unable to find any chaotic layout in the paper.
>
> c) It is not easy to reimplement the wFM layers, which is the core contribution of the paper.
>
> Ans: Since we used analytical expression for geodesics to compute wFM, for most of the manifolds encountered in the applications shown in this paper, the implementation of wFM is easy.
>
> d) Please see the anonymized link for the code:
>
> https://drive.google.com/open?id=1n18T7Ea-ides8NS-NEhnlG8PRK_DJM4e
>
> e) Anyway it fails to describe what kind of intrinsic metric they used for the specific manifold-valued data like SPD matrices and linear subspace.
>
> Ans: We used the canonical metric in both cases, i.e., GL-invariant canonical metric for SPD and the canonical metric for Grassmannian (which is the L2 norm of the principal angles between two subspaces).
>
> f) Is it \Gamma_{M_{n-1}}^{X_n} rather than \Gamma_{M_{n-1}}^{X_M} for Eq.(2)?
>
> Ans: M_n = \Gamma_{M_{n-1}}^{X_n}(.), since M_n lies on the geodesic between M_{n-1} and X_n. We do not think there is any X_M in this paper, so the reviewer probably misunderstood.
>
> g) For video classification, the paper only uses the moving MNIST, which is not a challenging dataset while there are plenty of large scale video datasets. In addition, the paper is expected to compare SOTA video classification methods.
>
> Ans: The primary goal of this paper is to introduce a theoretical framework that parallels CNNs but is suited for manifold-valued data sets. The paper introduces a novel technique to perform convolutions on general manifold valued data and also gave some proof-of-concept experiments.  Experiments with large scale video data sets will be one of our future aims and is currently not the focus of this paper. As mentioned above, the focus is to provide a novel intrinsic geometric framework for performing convolution operations on manifold-valued data sets and support the theory with proof of concept experiments.
>
> h) To learn the advantage of the proposed network over some related manifold networks like Huang et al., 2016, Huang & Van Gool 2017, it is necessary to evaluate them in the experiments.
>
> Ans: We already did a comparative experiment using Huang et al., 2017 on Moving MNIST data and the performance is as follows:30-60: 99.6%, 10-15: 50.4%, 10-15-20: 38.8%. Furthermore, each epoch took ~30.2 seconds and the total number of parameters is 110,000. We have tried with larger or smaller sized network but the performance is worse.
>
> i) Furthemore, it is also expected to compare more SOTA auto-encoder based reconstruction models like VAE [3], AAE [4] and WAE [5].
>
> Ans: In our auto encoder experiment, we did PCA in the encoding space, so if the encoding space follows a normal distribution (as in case of VAE), then doing PCA does not make sense as the encoding space is already decorrelated. This is the reason for not using a VAE type construction here.

---

### Official Review · AnonReviewer2 · 2018-11-03
**Interesting idea, however, more explanations needed.**

**Rating:** 5
**Confidence:** 3

**Review:**

This paper proposes a method to use weighted Frechet Mean (wFM) for the operation on Manifold valued data for CNN. The novel point is to view wFM as a convolutional layer. Overall, this paper is mathematically well written, however, how each theory improves CNN and the model used in experiments are not clear enough.

Pros
+ The use of wFM instead of a convolutional layer is an interesting idea.
+ This paper is mathematically well written.

Cos
- It is hard to understand how each theory presented in this paper helps to improve CNN. For example, the invariance to group operation. Some experimental results would help to understand the advantage of the group invariance.

- It is also unclear why the authors constructed the invariant last layer although the inputs of the last layer are invariant under group operations.

- In the introduction section, the authors raised the omnidirectional camera, diffusion magnetic resonance imaging, elastography as examples of manifold-valued data. However, experiments are limited to standard video sequences.

- It is unclear how to obtain the weights {w_i} of wFM by backpropagation.

- Since the contribution of this paper is to to use wFM instead of a convolutional layer, it is more interesting to visualize the weights {w_i}.

- More explanation needs for the model used for experiments. Especially in dimensional reduction experiments, I could not understand how each subspace is obtained and averaged. If each frame is a subspace, by averaging frames, the reconstruction would be blurred.

---

> ### Author Response · Authors · 2018-11-14
> **Thank the reviewer+answers to the comments**
>
> a)	It is hard to understand how each theory presented in this paper helps to improve CNN. For example, the invariance to group operation. Some experimental results would help to understand the advantage of the group invariance.
>
> Ans: Many classification problems should naturally exhibit group invariance, or more generally, group equivariance. For instance, an image of a cat remains an image of a cat regardless of any applied rigid or similarity transformation. More generally, whenever a NN must learn a function that exhibits group equivariance, it is highly desirable for the NN itself to exhibit group equivariance at each layer.   While the layers of our network are equivariant, we also design a special output layer that is invariant to group operations (just as in CNNs whose last layer is translation invariant) which is required for classification problems. This is precisely our contribution.
>
> In our moving MNIST experiment, we classify different moving patterns. But note that in any two sample videos from a same class, the digits started moving from random location. So, within a class, two paths correspond to two videos, related to one another by some affine transformation. In our representation, we used SPD matrices and the group that acts on the space of SPD matrices is the affine group. So, our network can classify two moving patterns if they are in the same orientation modulo any affine transformation. This is one example of the power of a group invariant network.
>
> b) It is also unclear why the authors constructed the invariant last layer although the inputs of the last layer are invariant under group operations.
>
> Ans:  The inputs to the last layer are not invariant to group operations admitted by the manifold on which the data reside, instead they are equivariant to these group operations. It seems like there is a misunderstanding in this context.  The output of each convolution is equivariant (not invariant) to the group action, and hence the input to the last layer is equivariant (not invariant). In order to make the output of the network invariant, we need a group invariant last layer. This last layer can be viewed as a substitute for a fully connected layer in traditional CNN’s (which is translation invariant).
>
> c) In the introduction section, the authors raised the omnidirectional camera, diffusion magnetic resonance imaging, elastography as examples of manifold-valued data. However, experiments are limited to standard video sequences.
>
> Ans: In the introduction section, we stated that there are two types of problems that can be consider here, (i) data that are manifold-valued and (ii) data that are samples of functions defined on manifolds. We stated example applications for both but clearly stated that the focus of this work is the problem stated in (i).   The example that this reviewer mentioned (such as, omnidirectional camera data) is a case that belongs to the problem in (ii) and is not the focus of our present paper. Other examples such as diffusion MRI, Elastography etc. do not have sufficient number of data sets that are publicly accessible. Further, they are specialized imaging techniques that are very expensive and public databases at most contain a few hundred scans, e.g., the Human Connectome Project data, which is insufficient amount of data for a deep network setting unless one uses a patch-based approach, which we will consider in the future.
>
> d) It is unclear how to obtain the weights {w_i} of wFM by backpropagation.
>
> Ans: Note that wFM is on the manifold and we provided a closed form recursive expression for its computation.  The weights {w_i} are however real valued and satisfy the convexity constraint. So, we used standard backpropagation to learn the {w_i}s.
>
>
> e) Since the contribution of this paper is to to use wFM instead of a convolutional layer, it is more interesting to visualize the weights {w_i}.
>
> Ans: For both the dimensionality reduction and the video classification experiments the weights reside in the temporal domain. Unlike for the regular CNN, where filter weights correspond to image features, our Temporal CNN structures are difficult to interpret by simple visualization since they capture temporal patterns.
>
>
> f) More explanation needs for the model used for experiments. Especially in dimensional reduction experiments, I could not understand how each subspace is obtained and averaged. If each frame is a subspace, by averaging frames, the reconstruction would be blurred.
>
> Ans: The k-dimensional subspaces spanned by the feature vectors correspond to the points on the Grassmannian. We take the wFM of these subspaces (which is a linear subspace) and then project the data onto this subspace. The projection is then passed onto the decoder. This is computationally a better substitute for doing PCA of the feature vectors directly (as shown in  Chakraborty et al CVPR17); since, for long videos, doing PCA on feature vectors is computationally infeasible.

---

> > ### Comment · AnonReviewer2 · 2018-11-30
> > **Thank you for the explanation.**
> >
> > Thank you for answering my questions. From the answers, several points of this paper become clear. However, my overall view of this paper has not been changed as the writing of this paper was not clear enough for me.
> >
> > Ans. b): ”The output of each convolution is equivalent (not invariant) …”
> > Thank you for pointing it out. There was a misunderstanding in this point; I was confusing equivalence and invariance.
> >
> > Ans. f): “We take wFM of theses subspaces ..”
> > I understand the proposed model averages linear subspaces on Grassmannian manifold. However, I still not understand how to obtain k-dimensional subspaces of feature vectors from video sequence for the input of wFM layer in the CNN model.

---

> > > ### Author Response · Authors · 2018-12-01
> > > **Response to second round of questions from Reviewer-2**
> > >
> > > (1) "However, my overall view of this paper has not been changed as the writing of this paper was not clear enough for me. "
> > >
> > > Ans: We are sorry that you feel the writing was unclear. Can you please point out any further clarifications that you need which we can provide in order to change this opinion of yours?
> > >
> > > We should point out that the content of our paper is primarily addressing non-Euclidean spaces and assumes readers with some background in non-Euclidean geometry as was expected from the readership of the "Spherical CNNs" paper in ICLR 2018 that was judged the best paper.
> > >
> > > (2) "I still not understand how to obtain k-dimensional subspaces of feature vectors from video sequence for the input of wFM layer in the CNN model."
> > >
> > > Ans: This was described in Section 3.2. the first paragraph in detail. We will now give a simplified explanation. Given a video, i.e., a time sequence of $N$ image frames, consider $k$ consecutive time frames and assemble them as column vectors of a matrix say $A$. The size of $A$ will then be $n\times k$. Now take the span of this matrix $A$ which will be a $k$-dimensional subspace in $\mathbf{R}^n$, hence will be a point on $Gr(k,n)$.
> > >
> > > Hope we have clarified all the issues that you have raised. We will be more than happy to clarify anymore that you might have.

---

### Meta-Review · Area_Chair1 · 2018-12-16

**Confidence:** 3
**Recommendation:** Reject

**Metareview:**

This manuscript proposes an extension of convolution operations for manifold-valued data. The primary contributions include the development and description of the approach and implementation and evaluation on real data.

The reviewers and AC expressed concern about the clarity of the presentation, particularly for a general ICLR audience. Though the contributions are primarily conceptual/theoretical, reviewers expressed concern about the breadth and quality of the presented experimental results. Some additional concerns related to missing proofs and details were addressed in the rebuttal.